# Understanding actions and challenges in protecting older people during covid-19 pandemic in indonesia: A qualitative study with female caregivers

Gregorius Abanit Asa[1,2]*, Nelsensius Klau Fauk[1,3], Melkianus Ratu[4], Elsa Dent[1], Paul Russell Ward[1]

1 Research Centre for Public Health, Equity and Human Flourishing (PHEHF), Torrens University Australia, Adelaide, South Australia, Australia, 2 Sanggar Belajar Alternatif (SALT), Atambua, Nusa Tenggara Timur, Indonesia, 3 Institute of Resource Governance and Social Change, Kupang, Indonesia, 4 Program Studi Keperawatan, Universitas Timor, Timor Tengah Utara, Nusa Tenggara Timur, Indonesia

* gorisasa@yahoo.com

**Data Availability Statement:** All relevant data are within the paper and its Supporting Information files.

## Abstract

COVID-19 has rapidly impacted societies on a global scale, with older people among the most affected. To care for older people living in their own homes, female family caregivers play a pivotal role. The current study aimed to explore the actions of female family caregivers and the challenges they faced in taking care of older people living at homes during the COVID-19 pandemic in Belu district, Indonesia. This qualitative study involved twenty female family caregivers, who were recruited using a combination of purposive and snowball sampling techniques. Findings were grouped into two main categories: (i) actions of female family caregivers in taking care of older adults during the COVID-19 pandemic. These included limiting both visitations of extended family members and older adults' activities outside homes; explaining the virus to older adults and controlling their access to news, social media and smartphones; providing nutrition, supplement and maintaining daily diets; and (ii) challenges they faced in taking care of older adults included excessive fear of contracting COVID-19 and possibility of transmitting it to older people; feeling stressed; tired and over-burdened. The study highlights the significant role family caregivers played to protect older people living at home. The findings can inform government to develop intervention programs that address and support the needs of both family caregivers and older people living at home.

## Introduction

COVID-19 has rapidly impacted the health and wellbeing of older people around the world. Older people, particularly those with chronic medical conditions, are reported to have the highest vulnerability to infections and fatalities from COVID-19 disease [1–3]. Evidence from the US shows that 75% of death cases due to COVID-19 occurred in adults aged over 65 [4],

**Funding:** The author(s) received no specific funding for this work.

**Competing interests:** The authors have declared that no competing interests exist.

which is in line with evidence from Italy where more than 80% of death-related COVID-19 came from people over 70 years old [5]. Similarly, in those above 60 years old in China, the mortality rate was 80% [6].

To care for older people, especially those living in their own homes during the COVID-19 pandemic, family caregivers have been the frontline workers [7]. They care for and protect older people with vulnerable health conditions such as age-related issues, chronic diseases, disability and mental health problems [8]. They are the invisible workforce of the health system [7]. While providing care for older adults at home, family caregivers concurrently face substantial pressure to also protect themselves from COVID-19 and prevent the transmission to older people they are caring for, which in turn can lead to various mental health issues among them [7, 9]. Previous studies in Spain, Italy and the United Kingdom showed that family caregivers for older adults with chronic illness experienced stress and anxiety during the COVID-10 outbreak [10, 11]. Family caregivers also face increased pressure in terms of personal, health and social care for older adults and other family members leading to negatively influencing their physical and psychological wellbeing [9, 11, 12]. Unlike caregivers in nursing homes, family caregivers work every day without appropriate COVID-related training, job descriptions or personal protective equipment (other than possibly face masks, which may not be appropriate or fitted well), which has been reported to add more burden to their daily responsibilities for other household chores during the COVID-19 pandemic [7]. As a consequence, family caregivers often feel overburdened, with a concern that some older adults living in their homes reported receiving less care and attention compared to those living in nursing homes [13].

Although family caregivers play important roles in care provision for older adults within families, to our knowledge there has been very limited evidence on family caregivers' actions and challenges they face in caring for older adults during the COVID-19 pandemic in resource-limited settings. Previous studies have mainly focused on understanding healthcare providers' perceptions and experiences of barriers to the provision of care and treatment for older people or older patients in nursing homes or healthcare facilities in developed countries [14–17]. This study aims to fill the gap by exploring what female family caregivers did and the challenges they faced in caring for older adults during the COVID-19 pandemic in Belu district, Indonesia. In many settings in developing countries, including Belu district, Indonesia, women (mothers, daughters, and daughters-in-law) have the responsibility to care for household chores, parents and other family members (children and husband) [18–22]. Understanding the roles of and challenges faced by female family caregivers can be useful in developing targeted interventions to address their needs and support them in the provision of care and protection for older people, especially in the current situation of COVID-19 pandemic. Moreover, the role of carers in Belu district is of fundamental importance, given that there are no nursing homes to assist in the care of the older population.

## Method

The study used consolidated for reporting qualitative studies (COREQ) to guide the report of the methods section of this study [23]. The COREQ checklist contains 32 required items (Fig 1) for explicit and comprehensive reporting of qualitative studies especially interviews and focus groups.

### Study setting

The study was conducted in Belu district, East Nusa Tenggara Province, Indonesia in February and March 2022. The district has a total population of 204,541 people that are distributed in 12 sub-districts [24]. The district shares the border with East Timor Country and has one public

| No | Item | Guide questions/description | Page |
|---|---|---|---|
| **Domain 1: Research team and reflexivity** | | | |
| Personal Characteristics | | | |
| 1. | Interviewer/facilitator | Which author/s conducted the interview or focus group? | 5-6 |
| 2 | Credentials | What were the researcher's credentials? E.g. PhD, MD | 5 |
| 3 | Occupation | What was their occupation at the time of the study? | 5 |
| 4 | Gender | Was the researcher male or female? | 5 |
| 5 | Experience and training | What experience or training did the researcher have? | 5 |
| Relationship with participants | | | |
| 6 | Relationship established | Was a relationship established prior to study commencement? | 6 |
| 7 | Participant knowledge of the interviewer | What did the participants know about the researcher? e.g. personal goals, reasons for doing the research | 5 |
| 8 | Interviewer characteristics | What characteristics were reported about the interviewer/facilitator? e.g. Bias, assumptions, reasons and interests in the research topic | 5-6 |
| **Domain 2: study design** | | | |
| Theoretical framework | | | |
| 9 | Methodological orientation and Theory | What methodological orientation was stated to underpin the study? e.g. grounded theory, discourse analysis, ethnography, phenomenology, content analysis | 5-6 |
| Participant selection | | | |
| 10 | Sampling | How were participants selected? e.g. purposive, convenience, consecutive, snowball | 5-6 |
| 11 | Method of approach | How were participants approached? e.g. face-to-face, telephone, mail, email | 5-6 |
| 12 | Sample size | How many participants were in the study? | 7 |
| 13 | Non-participation | How many people refused to participate or dropped out? Reasons? | 6 |
| Setting | | | |
| 14 | Setting of data collection | Where was the data collected? e.g. home, clinic, workplace | 5-6 |
| 15 | Presence of non-participants | Was anyone else present besides the participants and researchers? | 6 |
| 16 | Description of sample | What are the important characteristics of the sample? e.g. demographic data, date | 8-9 |
| Data collection | | | |
| 17 | Interview guide | Were questions, prompts, guides provided by the authors? Was it pilot tested? | 6 |
| 18 | Repeat interviews | Were repeat interviews carried out? If yes, how many? | 7 |
| 19 | Audio/visual recording | Did the research use audio or visual recording to collect the data? | 7 |
| 20 | Field notes | Were field notes made during and/or after the interview or focus group? | 6 |
| 21 | Duration | What was the duration of the interviews or focus group? | 6 |
| 22 | Data saturation | Was data saturation discussed? | 6 |
| 23 | Transcripts returned | Were transcripts returned to participants for comment and/or correction? | 6 |
| **Domain 3: analysis and findings** | | | |
| Data analysis | | | |
| 24 | Number of data coders | How many data coders coded the data? | 7 |
| 25 | Description of the coding tree | Did authors provide a description of the coding tree? | 7 |
| 26 | Derivation of themes | Were themes identified in advance or derived from the data? | 7 |
| 27 | Software | What software, if applicable, was used to manage the data? | 7 |
| 28 | Participant checking | Did participants provide feedback on the findings? | 6 |
| Reporting | | | |
| 29 | Quotations presented | Were participant quotations presented to illustrate the themes / findings? Was each quotation identified? e.g. participant number | 9-18 |
| 30 | Data and findings consistent | Was there consistency between the data presented and the findings? | 9-18 |
| 31 | Clarity of major themes | Were major themes clearly presented in the findings? | 9-18 |
| 32 | Clarity of minor themes | Is there a description of diverse cases or discussion of minor themes? | 18-21 |

**Fig 1. Consolidated criteria for reporting qualitative studies (COREQ): 32-item checklist.**

hospital known as MGR Gabriel Manek Hospital where patients with COVID-19 were quarantined, one army hospital, and two private hospitals known as Catholic Marianum hospital and Sitohusada Hospital. All the hospitals in study setting did not have specific services for older people during COVID-19 pandemic. Older people in the study setting are cared for predominantly by female family members, especially their daughters, in their private homes. This role

is influenced by cultural values putting expectation for women within the family to look after their parents.

In regards to the COVID-19 pandemic, as of 12th July, 2022 the current data show that there have been 6,112,986 confirmed COVID-19 cases in Indonesia. Of the total number, 5,935,845 people have fully recovered, 156,798 people died and 20,343 people are being treated [25, 26]. In East Nusa Tenggara, where Belu district is located, the current data as of 12th July 2022 report a total of 94,018 confirmed cases, of which 92,431 people have fully recovered, 1,524 people died and 63 people are being treated [25]. Belu is reported as one of the districts in East Nusa Tenggara province with high number of COVID-19 cases, accounting 536 [27] although it is a small district. Belu is selected because of small size, familiarity, and potential of undertaking the current study.

## Data collection

Data were collected using in-depth interviews: face-to-face using masks and via telephone and zoom. Participants were recruited using a combination of purposive and snowball sampling techniques. The process started when field researchers Gregorius Abanit Asa, MID and Melkianus Ratu, MHID (a nurse in the study setting) contacted several nurses who worked in a public hospital in the study setting and asked for help to disseminate the study information sheets to potential participants who accessed healthcare services at the hospital for older people they cared for. The nurses distributed the study information by posting it on the information board and WhatsApp groups. Potential participants who contacted the field researchers and confirmed to participate were recruited for an interview, which was conducted at a participant-researcher mutually agreed upon time. The initial participants who had been interviewed were also asked to disseminate the information to their eligible friends or colleagues or families who might be willing to take part in the study. The recruitment of the participants and interviews stopped when the research team felt that the data were rich enough to explain the topic being studied and there was an indication of data saturation as no new issues emerged from interviews with the last few participants [27]. Finally, 20 caregivers were interviewed. The inclusion criteria included (1) female family caregivers caring for an older person during the pandemic and (2) over 18 years old. All older adults they cared for were either their biological mother and/or father. Field notes were also taken during the interviews and integrated into each transcript during the transcription process. Only the researcher and participant were present in the interview room. The interviews were carried out in Indonesian, the primary language of the interviewers and participants. To protect the confidentiality of the participants, anonymity was ensured by using a letter and a number for each participant instead of their personal identifying detail. No participant required to withdraw or dropped out from the interview. The interviews were guided by several predetermined main questions and probing questions were developed during the interview [28]. Some examples of the main questions are "What actions did you take to protect older people or parents during COVID-19 pandemic? What challenges have you experienced when protecting older people during COVID-19 pandemic? What is your experience about older people's adherence to the actions taken to protect them from COVID-19? (S1 File). The decision about the questions was made through the process of formulation, discussion, and revision. Each participant was offered to correct or comment the transcription of the recording, but none asked to do so.

## Data analysis

Data were digitally recorded and transcribed. Content of the interview, emerging themes, coding and analysis were initially discussed by two of the authors (GAA and NKF) and further

discussed and refined by the remaining authors (MR, ED and PW). Analysis was conducted in Bahasa Indonesia before the quotes used in this manuscript were translated into English (by GA and NKF) to minimize the risk of losing semantic meaning. The translated quotes were then checked for clarity by other authors (PW and ED). Software package NVivo 12 was utilized for organizing data analysis. Analysis was guided by the five steps of qualitative data analysis introduced in Ritchie and Spencer's framework approach [29, 30]. The first was familiarization with data involving an iterative process of reading the transcripts, marking ideas and then making comments to search for meanings and patterns of ideas related to actions and challenges experienced by female family caregivers. The second stage was the identification of a thematic framework by writing down key points and concepts to identify themes. The third stage was indexing data by coding each transcript using the framework and analyzing codes to look for similar or redundant codes. Similar codes referring to the same theme were grouped together to reach a few overarching themes and sub-themes. The fourth step was creating a chart for themes and sub-themes by arranging indexed data related to the thematic framework. The fifth step was mapping and interpreting data as a whole.

## Ethical consideration

The study obtained the ethics approval from Health Research Ethics Committee, Duta Wacana Christian University, Yogyakarta, Indonesia (No. 1380/C.16/FK/2022). Each participant was informed about the aim of the study and there would be no consequences if they withdraw from the study without giving any reason. All participant signed and returned a written informed consent form via e-mail or WhatsApp a few days before the interviews. Each interview took 30–50 minutes and was recorded with the consent of the participants. Identification letter and number (e.g. R1, R2) was used for confidentiality purposes.

## Inclusivity in global research

Additional information regarding the ethical, cultural, and scientific considerations specific to inclusivity in global research can be found in the S2 File.

## Results

A total of 20 female family caregivers were included in the study. Of the 20 carers, 11 cared for one parent, and 9 cared for two parents. The mean age of carers and their older parents was 35.8 years and 67.3 years respectively. The details of the sociodemographic of participants and parents they cared for is presented in Table 1.

### Actions taken by female family caregivers in caring for older adults during the COVID-19 pandemic

**Limiting both visitations of extended families and older adults' activities outside homes.** The high level of vulnerability of older adults to COVID-19 transmission and its possible fatal consequences on their health and life required female caregivers to create strict rules to protect them from contracting the infection. One of the strict rules was the visitation limitation of all extended family members to meet older adults during the COVID-19 pandemic. Participants described they informed siblings and other family members not to visit their parents (older adults), and suggested them just drop items in front of the family house. These strict rules seemed to stem from their awareness of the possibility of COVID-19 transmission to older adults through extended family members, as reflected in the following narratives:

**Table 1. The sociodemographic profile of both female carers and their older parents receiving their care.**

| Female Caregivers | | | | | |
|---|---|---|---|---|---|
| Respondent No. | Age, years | Education | Occupation | Number of older people cared for | Age of F (Father) and M (Mother) |
| R1 | 34 | Senior high school | Cleaner | 2 | F = 65 |
| | | | | | M = 62 |
| R2 | 36 | Senior high school | Unemployment | 1 | M = 68 |
| R3 | 33 | Junior high school | Sales | 1 | F = 71 |
| R4 | 39 | Diploma | Nurse | 2 | F = 80 |
| | | | | | M = 76 |
| R5 | 39 | Undergraduate | Secretary | 1 | M = 72 |
| R6 | 32 | Senior high school | Entrepreneur (own a stall) | 1 | M = 75 |
| R7 | 36 | Senior high school | Admin | 1 | M = 65 |
| R8 | 29 | Undergraduate | Nurse | 2 | F = 67 |
| | | | | | M = 61 |
| R9 | 35 | Diploma | Teacher | 1 | M = 64 |
| R10 | 31 | Senior high school | Entrepreneur (own a stall) | 1 | M = 66 |
| R11 | 38 | Diploma | Nurse | 1 | M = 69 |
| R12 | 30 | Senior high school | Admin | 1 | M = 71 |
| R13 | 33 | Diploma | Nurse | 2 | F = 70 |
| | | | | | M = 66 |
| R14 | 36 | Undergraduate | Nurse | 2 | F = 68 |
| | | | | | M = 63 |
| R15 | 34 | Elementary school | Unemployment | 1 | F = 65 |
| R16 | 41 | Junior high school | Entrepreneur (own a stall) | 1 | M = 67 |
| R17 | 43 | Diploma | Early childhood teacher | 2 | F = 69 |
| | | | | | M = 64 |
| R18 | 37 | Undergraduate | Unemployment | 2 | F = 71 |
| | | | | | M = 63 |
| R19 | 42 | Diploma | Nurse | 2 | F = 65 |
| | | | | | M = 64 |
| R20 | 37 | Senior high school | Sales | 2 | F = 64 |
| | | | | | M = 62 |

"We have a big family and we used to gather together in this home before the COVID-19 pandemic. We can spontaneously gather without any plan before. They just came home. You know, when the first COVID-19 case was detected in this district, I messaged my uncles, aunties and grandchildren to not visit our home as our mother was already old and sick. I am aware of the high possibility of COVID transmission to our mother through family members who come to see her" (R10: 31 years old entrepreneur).

"Usually, my sister stopped by our house to drop food or fruits almost every morning to our parents when she was on the way to work. But everything changed since we had COVID-19 cases in the district. I asked her to just drop the food in front of the door and not go into the house to meet our mother or father in their room" (R19: 42 years old nurse).

Another strict action taken by female caregivers to protect older adults from COVID-19 transmission was limiting their activities outside the house. This included limiting their involvement in activities within the community and not allowing them to visit neighbors or go to market for grocery shopping. Participants also reported that they were happy as their

parents complied with their suggestions or instructions, however, they were not sure about the feelings or emotional state experienced by older adults during that period or due to restrictions imposed on them. The following narratives illustrated these assertations:

> "I was really strict with my mother and band her from going outside the house. I had to do that because my mother was still independent although she was in her 70s. The good news was that she listened to me because I am the one who lives with her and helps her every single day but to be honest, I did not know how she felt. I didn't ask her and she did not say anything to me either" (R12: 32 years old admin).

> "My mother wanted to go to the market during COVID-19. She said she would wear masks and wash her hands. However, I did not want. I told her that my other siblings and I didn't agree. Practically, I banned her from going outside our house. Probably she did not like it. . ." (R5: 39 years old secretary).

**Explaining about the virus to older adults and controlling news, social media and smartphones.**   Knowledge about COVID-19 infection and its negative impact such as death was another factor that underpinned action taken by female caregivers to protect their older adults from the infection. Participants reported repeatedly explaining and reminding older adults about adherence to COVID-19 protocols such as washing hands, wearing facemasks, avoiding crowded places and social distancing. These were done with the expectation that such knowledge or information could help older adults comply with or perform preventive behaviors as described in the following quotes:

> "I told my mother and father that the virus is very dangerous and could cause death immediately in particular for older people like them. You know, I had to repeat the same sentences every day because they might forget what I said before. I also told and explained almost every day about the importance of washing hands, wearing masks, keeping physical distance and so on to protect them and all of us in the family" (R8: 29 years old nurse).

> "I told to my mother that COVID-19 spreads very quickly from one person to another and we don't know who the carrier of the virus is. The carrier could be my brother, my sister, her grandchildren and could be myself. My mother understood that but I had to say that several times in a day" (R14: 36 years old nurse).

Several participants reported that they controlled the news their parents could watch and read. For example, some did not want their parents to watch too much news about COVID-19 and switched TV channels to dramas. Not allowing older adults to have smart-phones through which they may access COVID-19-related information that could cause fear, stress or anxiety to them was another action undertaken by the caregivers to protect the older adults they cared for:

> "I used to control the TV program my parents watched. I did not want them to watch too much news on COVID-19 and the toll death due to the virus. Sometimes they watched the news about COVID-19 but not that often. I always chose the channel that had a lot of drama" (R13: 33 years old nurse).

> "My parents had old mobile phones that they could use only for texting and calling. I did not want them to have smart-phones that have WhatsApp application and internet connections. This prevented them from following or updating circulated news through WhatsApp

or other applications or accessing information about COVID-19 cases or the number of people who died from COVID-19. If they followed the news from social media, they might be more stressed with negative news such as death toll and the increased number of COVID-19 cases" (R17: 43 years old teacher).

**Providing nutrition, supplement and keeping daily diets.** Female caregivers were also aware of the importance of a nutritious food supply to support the immune system against infection. Such an awareness was put into practice through the provision of nutritious food for older adults to help them support their body immune and help them stay healthy. The provision of nutritious foods for older adults was considered necessary because older adults spent more time at home and lacked physical activities as illustrated in the quotes below:

"Before COVID-19 exists, I never thought of providing regular supplements or fruits to my mother and father. But now, I thought I needed to provide for them. Maybe not every day, but at least once every three days to support their immune system" (R4: 39 years old nurse).

"Parents did not walk a lot during COVID-19, lack of exercise. They spent all day at home. This caused their immune system to decrease. Therefore, it is important to support them with healthy food, with supplements. I gave them supplements almost every day" (R19: 42 years old nurse).

Although the provision of nutritious food was considered important for older adults, most participants acknowledged that they did not have extra money to buy fruits or supplements every day for their parents (older adults). As a consequence, some participants described that they did not regularly provide nutritious food for parents during the pandemic. Difficult financial condition which seemed to be stemming from unemployment and lack of income was the main reason for the unaffordability of nutritious food for older adults they cared for:

"Sometimes I gave my parents vitamin C which I got from the pharmacy. To be honest, vitamins, fruits, and other supplements were not cheap. I could not give them every day. If they had symptoms then I would provide them" (R20: 37 years old sales).

"I knew it was suggested on TV and news about nutritious foods. However, I only prepared their normal food every day. You know, nutritional food and supplements were not cheap. To be honest, we did not have extra money to buy different food (nutritional food) and supplements. We already struggle with daily needs. I just protected them (parents) to not getting infected with the virus. I am just a housewife and do not have a paid job or income" (R1: 34 years old cleaner).

## Challenges faced by female family caregivers in providing care and protection for older adults during the COVID-19 pandemic

**Excessive fear of contracting COVID and the possibility of transmitting it to older people.** Excessive fear among female caregivers due to the awareness of the high possibility of contracting COVID-19 infection, passing it to older adults and experiencing its negative impact on themselves and the older adults they cared for was a major challenge facing them. The perceived challenging situation that would be faced by older adults if participants had to be in quarantine due to contracting the infection was also another supporting factor for such

fear. Similarly, continuous news and information about the length of survival of the coronavirus up to several days on the surface of metal and plastic supported such fear the caregivers felt:

> "The biggest challenge for me was if I got infected with the virus and I could pass it to my mother and father who had comorbidities. Or if my parents were lucky (tested negative) and I had to be in quarantine for several days, I did not know who would help them" (R4: 39 years old nurse).

> "Every day I was just at my small stall because this is my small business. I felt a bit safe here. However, I might get infected by the virus because according to the news the virus could survive several days on the surface of plastics and metal. You know, most pieces of stuff in my stall were covered by plastics. If I was COVID-19 positive, then the others my home might be positive and this scared me" (R16: 41 years old entrepreneur).

Some participants also reported the same fear of contracting COVID-19 infection due to their big role in their families, including taking care of older adults and other household chores. Thus, a diagnosis of COVID-19 in them was considered to add the burden to themselves and the older adults they cared for:

> "My responsibility at home is not only caring for my mother. I had to do other chores such as cooking, cleaning and shopping. Imagine if I got infected. It's would be harder for my mother and me" (R10: 31 years old entrepreneur).

**Feeling stressed.**   Feeling stressed was another challenge female caregivers faced while being responsible for caring for and protecting their older adults during the COVID-19 pandemic. This was supported by several reasons, including older people's doubt about the existence and severity of the impact of COVID-19 and the number of COVID-19-related death cases. Similarly, they disbelieved in the deaths of healthy people several days after contracting the virus. Some female caregivers acknowledge that such doubt had an implication on the behaviors of some older adults reflected in their underestimation of health protocols and involvement in communal activities with other people as described in the quotes below:

> "I sometimes feel stressed looking after my father during the COVID-19 pandemic as he does not believe that the virus spreads quickly and causes death instantly. He once said it is impossible that those who looked healthy then died several days after contracting the virus. My father believes that this is only media propaganda" (R1: 34 years old nurse).

> "I was a bit stressed as he (my father) was brave enough to go out of our house and gather with his friends engaging in cockfighting gambling. This activity was done secretly and a lot of males were there. When I asked him not to do that activity during the pandemic, he replied that all people in the cockfighting arena were healthy and no need to be worried" (R8: 29 years old nurse).

Feeling stressed facing female caregivers was also facilitated by older people's attitude to only listen to instructions coming from certain children at home. In this study, parents tended to only listen to the oldest son and oldest daughter leading some female family caregivers to a difficult situation to control parents (older adults) when they were not at home. The following statements describe such challenging experiences:

"My mother was a very active woman when she was young. So, it was quite challenging to prevent her from going outside the house. If she noticed no one watched or supervised her at home, she could easily go to the market or visit neighbors. She also sometimes did not listen to my youngest sister. So, to control her activities, I asked my sister to text or call me if my mother sneaked out from home" (R5: 39 years old secretary).

"Sometimes I found it difficult with my mother because she tended to only listen to my oldest brother. So, after I banned her not to go outside the house, I told her that my oldest brother will come if she did not listen to me" (R12: 30 years old admin).

**Feeling tired and overburdened.**    Caring for older adults, during the COVID-19 pandemic, imposed a significant burden on female caregivers. Most participants acknowledged feeling tired and overburdened during the COVID-19 pandemic due to extra tasks and responsibilities. For example, in addition to work, they had to wipe almost all the surfaces of the stuff in their houses and monitored the movement of their parents and controlled people who visited them in the house. Stories of a few participants illustrate such action:

"Prior to this pandemic I just went to work and did not think too much about my parents because they are still active and independent. But now, you know, it's totally different. I have to make sure the house and surrounding are clean. I wipe all pieces of stuff covered by plastics in my small stall every day and I make sure the house is clean. So, now I have more work. I feel tired but I have to do that" (R19: 42 years old nurse).

"Monitoring parents who remained active and independent is not easy because they do not want to be controlled or monitored. If I do not monitor them, they might go to market or engage with neighbors. In addition to thinking about my work, I also have to control people who visit our home, making sure they wash their hands before they enter the house, wear masks, and keep physical distance from my parents. Of course, I have too much work but I cannot complain because all this for our safety" (R4: 39 years old nurse).

## Discussion

In the present study, we explored the actions of female family caregivers and the challenges they faced in taking care of and protecting older people from COVID-19 in Belu district, Indonesia. Our study suggests that female family caregivers were aware of the severe impacts of COVID-19, and knowledgeable about the older population as a vulnerable group affected by the disease with high fatalities. This knowledge was reflected in a range of actions undertaken to limit interactions of older adults with extended family members, neighbors and other community members to prevent their exposure to COVID-19, which are congruent with COVID-19 preventive protocols [31, 32] and previous findings reported elsewhere [33, 34]. The actions of the female caregivers seem to also reflect the social behavior theory suggesting that infection may determine behaviors of individuals or communities and alter or reduce social connectivity to prevent transmission [35–37]. Our findings also show the perceived importance of COVID-19-related health and social media literacy for older adults, which led to carer provision of the necessary information about COVID-19 and the control of social media and smart-phones use among older people. A possible explanation for such actions was the perceived lack of critical thinking among older adults to analyze the overload COVID-19-related news or information

which could, in turn, result in negative consequences to their emotional states such as anxiety, worry, stress, depression and sadness, as reported in previous studies [38–41].

The study also suggests a dilemmatic situation facing the caregivers with regards to the importance of a healthy diet for older adults to boost their body immune system during the COVID-19 pandemic and their difficult financial condition. Their knowledge or understanding of the importance of a healthy diet or good nutritional status for older people is consistent with the reports of previous studies [42–44], however, the findings indicate that the caregivers' economic or financial hardships prevented the translation of the knowledge into practices. The findings supported previous studies which have reported that COVID-19 exacerbates the already existing financial stress, and has led many people including older adults to precarious situations in regard to food security [44] and the inability to afford a healthy diet [45, 46].

In line with previous findings [9, 10, 47, 48], the current findings suggest that female family caregivers experienced challenges in caring for and protecting older adults during the COVID-19 pandemic. Such challenges were manifested in mental health issues such as over fear, stress, and feeling tired and overburdened. The thoughts about the possibility of being infected, quarantined, and transmitting to older people was reported as the major perceived contributing factors for such fear among the caregivers. Such excessive fear seemed also to be supported by news about characteristics of the virus survival on the surface of certain materials [49]. This characteristic was reported to increase the workload for family caregivers, with the additional chore of needing to frequently clean commonly touched surfaces around the household–a finding consistent with previous studies [50, 51]. Furthermore, the findings showed that the caregivers faced psychological distress due to the older adults' skepticism of COVID-19 and its severe impact. This skepticism then translated into behavior and practice as evidenced by non-adherence to health protocols, which has also been found in previous research [52].

Similarly, female family caregivers felt tired and overburdened as a result of having extra responsibilities for monitoring older adults' movements and controlling people who visited their homes. This burden could have likely been exacerbated by the absence of any paid domestic workers during COVID-19 lockdown periods. Furthermore, in many parts of Asia, it is a common expectation by older adults that their children will look after them when they are older–pressure to do so may have contributed to some of the mental health issues felt by the female caregivers in our study. For instance, several recent studies have highlighted the link between caregiver burden and the level of care that children are obliged to provide for their older parents–this is true across all cultures, particularly in Asian countries [53–55]. Future research should focus on direct population-wide strategies to reduce the caregiver burden for those looking after their older relatives, including the promotion of less gendered caregiving and the utilization of non-family caregivers, such as paid assistance [55]. An additional strategy to potentially alleviate carer burden is to involve older adults in their own for, including asking what their preferences are. For example, recent research from Australia showed that when an older adult participated in their own healthcare, trust and report with their carer was strengthened [56].

Of important note, the average age of the older adults in our study receiving the care was 67.3 years; the average lifespan in Indonesia is currently 69.4 years for males, and 73.3 years for females [57]. Moreover, life expectancy has experienced unprecedented growth over the last four decades in Indonesia, with life expectancy at 60 years growing from 13.74 years to 18.28 years during this time [58]. As life expectancy continues to rise globally in both high- and low-middle income countries, it will become increasingly important in the future to alleviate the pressure from already overburdened female caregivers, who will need to provide many

additional years of care for their older relatives. Thus, it is urgent to devise public health strategies to assist in the dignified care of older adults.

## Study limitations and strengths

There are several limitations of the study. First, researchers could not see visual cues or expressions of the participants when conducting the telephone interview. Second, the study only explored the perspectives of female family caregivers and did not explore the view of other family members, who may have provided different stories regarding their actions and challenges facing them in taking care of and protecting older adults during the COVID-19 pandemic. Third, female caregivers may not have disclosed all information to the male field researchers. Fourth, it is not known how many caregivers in our study usually had additional part-time or full-time paid domestic workers providing care for their older parents before the COVID-19 pandemic. Finally, the study was carried out in only one district reflecting the unique conditions of participants in the study setting, and may not be generalizable to other settings. The strength of the study is that it is first investigation of the roles of and challenges faced by female family caregivers of older adults in Indonesia during the COVID-19 pandemic. Thus, our findings have important implications for the government to allocate resources that can support families with older adults, especially during the challenging situation of COVID-19 pandemic.

## Conclusion

The study presents female family caregivers' actions and challenges they experienced in taking care of older adults during the COVID-19 pandemic. The actions were limiting both visitations of extended families and older adults' activities outside homes; explaining the virus to older adults and controlling news, social media and smartphones; providing nutrition, supplement and maintaining daily diets. Female family caregivers also experienced challenges reflected in over fear of being infected and infecting older adults, feeling stressed, tired and overburdened. The study enhances our understanding of how family caregivers play crucial roles to the best of their ability at the invisible level of the health system to protect older adults at home. The results of the study can contribute to developing intervention programs such as providing food support and nutritional supplements for female family caregivers and older people living at home in poor or limited resource settings. The results of the study can contribute to developing intervention programs that address and support the needs of both female family caregivers and older people living at home in poor or limited resource settings. Large-scale studies to understand the dynamic within families taking of older adults in their private homes in Indonesia are recommended as their findings can better inform policy and practice.

## Supporting information

**S1 File. Interview guide.**
(DOCX)

**S2 File. Inclusivity in global research.**
(DOCX)

## Author Contributions

**Conceptualization:** Gregorius Abanit Asa, Nelsensius Klau Fauk.

**Formal analysis:** Gregorius Abanit Asa, Nelsensius Klau Fauk.

**Investigation:** Gregorius Abanit Asa, Melkianus Ratu.

**Methodology:** Gregorius Abanit Asa, Nelsensius Klau Fauk, Elsa Dent, Paul Russell Ward.

**Project administration:** Gregorius Abanit Asa, Nelsensius Klau Fauk, Melkianus Ratu.

**Writing – original draft:** Gregorius Abanit Asa.

**Writing – review & editing:** Gregorius Abanit Asa, Nelsensius Klau Fauk, Elsa Dent, Paul Russell Ward.

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
