## [Decision Letter · Decision Letter 0]

30 Jan 2023

PONE-D-22-22781UNDERSTANDING ACTIONS TAKEN BY FEMALE FAMILY CAREGIVERS AND CHALLENGES THEY FACED IN CARING FOR OLDER PEOPLE DURING COVID-19 PANDEMIC IN BELU DISTRICT, INDONESIA: A QUALITATIVE STUDYPLOS ONE

Dear Dr. Asa,

Thank you for submitting your manuscript to PLOS ONE. After careful consideration, we feel that it has merit but does not fully meet PLOS ONE’s publication criteria as it currently stands. Therefore, we invite you to submit a revised version of the manuscript that addresses the points raised during the review process.

We look forward to receiving your revised manuscript.

Kind regards,

Farooq Ahmed, PhD

Academic Editor

PLOS ONE

and https://journals.plos.org/plosone/s/file?id=ba62/PLOSOne_formatting_sample_title_authors_affiliations.pdf.

3. Please include a complete copy of PLOS’ questionnaire on inclusivity in global research in your revised manuscript. Our policy for research in this area aims to improve transparency in the reporting of research performed outside of researchers’ own country or community. The policy applies to researchers who have travelled to a different country to conduct research, research with Indigenous populations or their lands, and research on cultural artefacts. The questionnaire can also be requested at the journal’s discretion for any other submissions, even if these conditions are not met.  Please find more information on the policy and a link to download a blank copy of the questionnaire here: https://journals.plos.org/plosone/s/best-practices-in-research-reporting. Please upload a completed version of your questionnaire as Supporting Information when you resubmit your manuscript.

Reviewers' comments:

Reviewer's Responses to Questions

**Comments to the Author**

1. Is the manuscript technically sound, and do the data support the conclusions?

Reviewer #1: Yes

Reviewer #2: Yes

2. Has the statistical analysis been performed appropriately and rigorously? 

Reviewer #1: N/A

Reviewer #2: N/A

3. Have the authors made all data underlying the findings in their manuscript fully available?

Reviewer #1: Yes

Reviewer #2: Yes

4. Is the manuscript presented in an intelligible fashion and written in standard English?

Reviewer #1: Yes

Reviewer #2: Yes

5. Review Comments to the Author

Reviewer #1: Dear authors,

Thanks for the opportunity to review this paper. This paper presents very important findings about older people who are vulnerable to COVID-19 infection and can inform policy and programs to support this population.

Overall, the manuscript is well-written. All parts have been explained very well by the authors. The gap in knowledge is identified, which justifies the study. I have a few suggestions that hopefully help the authors for further improvement of their manuscript.

1. They need to add a brief explanation or justification about why they chose to conduct the study in Belu.

2. Data saturation: there seems to be a repetition of information about data saturation as it is mentioned in two places.

3. Did face-to-face interviews follow COVID-19 prevention protocols? If yes, please make it clear in the methods section.

4. There is a supplementary file of COREQ checklist but you haven’t mentioned it in the text, please report it stating that you follow the guideline and why.

Reviewer #2: The study explores challenges faced by female caregivers in protecting older adult family members during the COVID-19 pandemic. The findings are very interesting and relevant and can be considered for publication.

Some comments to be considered by the authors:

I am wondering whether the authors can provide further explanation about why they included only female caregivers. The authors stated, “Older people in the study setting are cared for predominantly by female family members, especially their daughters, in their private homes”. Does it have something to do with culture or religion? Or something else? Also, it would be good to justify the selection of the study setting.

Do the hospitals or healthcare facilities in the study setting have specific services for older people during the COVID-19 pandemic?

It would be helpful for the readers if the authors can provide some examples of the main research questions used to explore the topic.

What are the implications of your findings for the health department or the government in the study setting? The authors mentioned intervention programs to support both female caregivers and older people. Can you propose any appropriate intervention programs that may have been implemented in other settings? Food support, nutritional supplement support, etc….

6. PLOS authors have the option to publish the peer review history of their article (what does this mean?). If published, this will include your full peer review and any attached files.

Reviewer #1: No

Reviewer #2: No

---

## [Author Response · Author response to Decision Letter 0]

6 Feb 2023

Response to Reviewers

Dear Editor,

Thank you very much for considering our manuscript " UNDERSTANDING ACTIONS TAKEN BY FEMALE FAMILY CAREGIVERS AND CHALLENGES THEY FACED IN CARING FOR OLDER PEOPLE DURING COVID-19 PANDEMIC IN BELU DISTRICT, INDONESIA: A QUALITATIVE STUDY" to be published in POLOS ONE Journal. Herewith we submit the revised version of our paper. 

On behalf of all authors,

Sincerely,

Gregorius Abanit Asa

RESPONSE TO THE JOURNAL

Comment

Please ensure that you refer to Table 1 in your text as, if accepted, production will need this reference to link the reader to the Table.

Response

Table 1 has been referred in the text (in the result section prior to the table)

Comment

Please upload a copy of Figure 1 which you refer to in your text. Or if the figure is no longer to be included as part of the submission please remove all reference to it within the text.

Response

The copy of Figure 1 has been uploaded.

Comment

Response

The manuscript follows the PLOS ONE’s style requirements.

Comment

Please provide additional details regarding participant consent. In the ethics statement in the Methods and online submission information, please ensure that you have specified (1) whether consent was informed and (2) what type you obtained (for instance, written or verbal, and if verbal, how it was documented and witnessed).

Response

The study obtained the ethics approval from Health Research Ethics Committee, Duta Wacana Christian University, Yogyakarta, Indonesia (No. 1380/C.16/FK/2022). Each participant was informed about the aim of the study and there would be no consequences if they withdraw from the study without giving any reason. All participant signed and returned a written informed consent form via e-mail or WhatsApp a few days before the interviews. Each interview took 30-50 minutes and was recorded with the consent of the participants. Identification letter and number (e.g. R1, R2) was used for confidentiality purposes. 

Comment 

Please include a complete copy of PLOS’ questionnaire on inclusivity in global research in your revised manuscript. Our policy for research in this area aims to improve transparency in the reporting of research performed outside of researchers’ own country or community. The policy applies to researchers who have travelled to a different country to conduct research, research with Indigenous populations or their lands, and research on cultural artefacts. The questionnaire can also be requested at the journal’s discretion for any other submissions, even if these conditions are not met. Please find more information on the policy and a link to download a blank copy of the questionnaire here: https://journals.plos.org/plosone/s/best-practices-in-research-reporting. Please upload a completed version of your questionnaire as Supporting Information when you resubmit your manuscript.

Response

Inclusivity in global research form has been filled (see supporting Information file)

RESPONSE TO REVIEWERS

REVIEWER 1

Comment

They need to add a brief explanation or justification about why they choose to conduct the study in Belu.

Response

Belu is reported as one of the districts in East Nusa Tenggara province with high number of COVID-19 cases, accounting 536 although it is a small district. Belu is selected because of small size, familiarity, and potential of undertaking the current study. 

Comment 

Data saturation: there seems to be a repetition of information about data saturation as it is mentioned in two places.

Response

This sentence “The recruitment stopped when the authors felt the data saturation had been achieved” is removed because it is a repetition. 

Comment

Did face-to-face interviews follow COVID-19 prevention protocols? If yes, please make it clear in the methods section.

Response

Data were collected using in-depth interviews: face-to-face using masks and via telephone and zoom. 

Comment

There is a supplementary file of COREQ checklist but you haven’t mentioned it in the text, please report it stating that you follow the guideline and why.

Comment

The study used consolidated for reporting qualitative studies (COREQ) to guide the report of the methods section of this study. The COREQ checklist contains 32 required items (Fig.1) for explicit and comprehensive reporting of qualitative studies especially interviews and focus groups. 

REVIEWER 2

Comment

I am wondering whether the authors can provide further explanation about why they included only female caregivers. The authors stated, “Older people in the study setting are cared for predominantly by female family members, especially their daughters, in their private homes”. Does it have something to do with culture or religion? Or something else? Also, it would be good to justify the selection of the study setting.

Response

Older people in the study setting are cared for predominantly by female family members, especially their daughters, in their private homes. This role is influenced by cultural values putting expectation to women within the family to look after their parents. 

Comment

Do the hospital or healthcare facilities in the study setting have specific services for older people during the COVID-19 pandemic?

Response

All the hospitals in study setting did not have specific services for older people during COVID-19 pandemic. 

Comment

It would be helpful for the readers if the authors can provide some examples of the main research questions used to explore the topic.

Response

The interviews were guided by several predetermined main questions and probing questions were developed during the interview. Some examples of the main questions are “What actions did you take to protect older people or parents during COVID-19 pandemic? What challenges have you experienced when protecting older people during COVID-19 pandemic? What is your experience about older people’s adherence to the actions taken to protect them from COVID-19? The decision about the questions was made through the process of formulation, discussion, and revision.

Comment

What are the implications of your findings for the health department or the government in the study setting? The authors mentioned intervention programs to support both female caregivers and older people. Can you propose any appropriate intervention programs that may have been implemented in other settings? Food support, nutritional supplement support, etc….

Response

The results of the study can contribute to developing intervention programs such as providing food support and nutritional supplements for female family caregivers and older people living at home in poor or limited resource settings.

---

## [Editor Report · Decision Letter 1]

6 Mar 2023

UNDERSTANDING ACTIONS TAKEN BY FEMALE FAMILY CAREGIVERS AND CHALLENGES THEY FACED IN CARING FOR OLDER PEOPLE DURING COVID-19 PANDEMIC IN BELU DISTRICT, INDONESIA: A QUALITATIVE STUDY

PONE-D-22-22781R1

Dear Dr. Gregorius Abanit Asa,

We’re pleased to inform you that your manuscript has been judged scientifically suitable for publication and will be formally accepted for publication once it meets all outstanding technical requirements.

Kind regards,

Farooq Ahmed, PhD

Academic Editor

PLOS ONE

---

## [Editor Report · Acceptance letter]

9 Mar 2023

PONE-D-22-22781R1 

UNDERSTANDING ACTIONS AND CHALLENGES IN PROTECTING OLDER PEOPLE DURING COVID-19 PANDEMIC IN INDONESIA: A QUALITATIVE STUDY WITH FEMALE CAREGIVERS 

Dear Dr. Asa:

I'm pleased to inform you that your manuscript has been deemed suitable for publication in PLOS ONE. Congratulations! Your manuscript is now with our production department. 

Kind regards, 

on behalf of

Dr. Farooq Ahmed 

Academic Editor

PLOS ONE